# Traffic Noise Assessment Using Intelligent Acoustic Sensors (Traffic Ear) and Vehicle Telematics Data

**DOI:** 10.3390/s23156964

**Published:** 2023-08-05

**Authors:** Omid Ghaffarpasand, Anwar Almojarkesh, Sophie Morris, Elizabeth Stephens, Alaa Chalabi, Usamah Almojarkesh, Zenah Almojarkesh, Francis D. Pope

**Affiliations:** 1School of Geography, Earth, and Environmental Sciences, University of Birmingham, Birmingham B15 2TT, UK; 2Innovation Factory Limited, Birmingham B7 4BP, UK; almojarkesh@innovfactory.com (A.A.);; 3Sandwell Metropolitan Borough Council, Sandwell B69 3DE, UK

**Keywords:** traffic noise, acoustic sensors, machine learning, telematics data, road transport

## Abstract

Here, we introduce Traffic Ear, an acoustic sensor pack that determines the engine noise of each passing vehicle without interrupting traffic flow. The device consists of an array of microphones combined with a computer vision camera. The class and speed of passing vehicles were estimated using sound wave analysis, image processing, and machine learning algorithms. We compared the traffic composition estimated with the Traffic Ear sensor with that recorded using an automatic number plate recognition (ANPR) camera and found a high level of agreement between the two approaches for determining the vehicle type and fuel, with uncertainties of 1–4%. We also developed a new bottom-up assessment approach that used the noise analysis provided by the Traffic Ear sensor along with the extensively detailed urban mobility maps that were produced using the geospatial and temporal mapping of urban mobility (GeoSTMUM) approach. It was applied to vehicles travelling on roads in the West Midlands region of the UK. The results showed that the reduction in traffic engine noise over the whole of the study road was over 8% during rush hours, while the weekday–weekend effect had a deterioration effect of almost half. Traffic noise factors (dB/m) on a per-vehicle basis were almost always higher on motorways compared the other roads studied.

## 1. Introduction

Road transport represents a key pillar for economic growth, social welfare, and sustainable development. For example, in 2019, the road transport and storage services sector (including postal and courier activities) in the European Union (EU) countries employed over ten million people, equating to about 5.3% of the total EU workforce [1]. However, it is a leading source of a variety of undesirable and unsustainable outcomes. The transportation sector consumes over a quarter of the world’s energy annually [2]; hence, it is one of the leading causes of global climate change and urban air pollution at a local scale [3].

Road transport is also a major source of noise pollution in the urban environment. Reports from Spain [4], Ghana [5], and China [6], have all shown that traffic is the main cause of noise in urban environments. The serious impacts of noise pollution on human health have been evidenced by a wide body of research. For example, Bao et al. highlighted the strong correlation between noise exposure and behavioural problems in children in China [6]. Sørensen et al. studied the potential links between noise exposure and the risk factors for type 2 diabetes, which showed that high levels of noise pollution were associated with an increased risk of diabetes [7]. We refer the reader to systematic reviews for more information on the health effects of noise exposure, such as the studies published by the authors of [8,9].

Noise maps are currently the major policy instruments that enable noise hotspot analysis for environmental noise management and planning [10]. They help policymakers and urban managers in decision making on environmental noise regulations. One of the first urban noise maps was made for the EU countries as a consequence of the Environmental Noise Directive (END) known as Directive 2002/49/EC [11]. This Directive legally obligated EU countries to develop strategic noise maps and their corresponding action plans every five years [12]. In the UK, the latest noise strategy map was developed by the Department for Environment, Food, and Rural Affairs (DEFRA) [13]. UK noise maps have been historically developed using computer modelling that incorporates information, such as traffic flow and vehicle type, whereas no actual measurements are used in the production of these strategic maps. Therefore, they do not represent a reliable account of noise across road transport in the UK. Strategic noise maps do not usually attempt to cover all environmental noise sources. For example, traffic on minor roads is often not considered due to the lack of road data [14]. Temporal variations are also often neglected in strategic road maps; hence, most of them are annually averaged [15].

Traffic noise maps typically consider vehicles with engines operating on the road to be the sole source of noise within the model. They use daily traffic flow as a non-acoustic proxy to estimate noise emission from the roads [16,17,18]. In addition to the traffic flow, the average speed of the vehicles moving over the roads, also referred to as the traffic flow speed, is also often used in developing noise maps.

Traffic noise maps are usually developed through either measurement-based or model-based methods [10]. Within measurement-based models, environmental noise is monitored through densely distributed acoustic sensors; the noise map can be developed using various interpolation techniques. For example, Tsai et al. used the spatial interpolation method along with their data collected from over 345 acoustic monitoring sensors to develop the noise maps of the city of Tainan, Taiwan [19]. The establishment of such a network of densely distributed acoustic sensors is typically quite expensive, especially for wide areas and regions. Moreover, there are debates on the quality of outputs generated through interpolation methods, see for example the study published by the authors of [20]. Meanwhile, the model-based methodology to generate noise maps relies on noise prediction models to estimate and develop noise maps, see for example the study published by the authors of [21].

Among the model-based developed traffic noise maps, Zamon et al. assessed subset roads in homogenous clusters and attributed certain levels of traffic noise to each cluster according to their main traffic flow and speed [22]. However, such road classification neither reflects the real-world status of roads nor noise emissions across the road network. Traffic noise maps have, for example, been developed for many cities in China by several previous investigators, such as the ones published by the authors of [23,24,25]. Along with the traditional measurements and model-based methods, machine learning approaches have recently come to the forefront of traffic noise prediction. For example, Adulaimi et al. used a land use regression (LUR) model based on machine learning to determine traffic noise from the surrounding noise in Shah Alam, Malaysia [26]. They then considered several involved factors, such as traffic lights, intersections, road toll gates, gas stations, and public transportation infrastructures to develop the traffic noise map of the studied area. Yin et al. analysed the functionality of four machine learning models: linear regression, random forest, extreme gradient boosting, and a neural network in predicting the traffic noise, and found the best results in validation tests for the extreme gradient boosting model [27]. Fallah-Shorshani et al. assessed the traditional model-based methods with the recently developed machine learning approaches in developing the traffic noise maps [28]. They uncovered a significantly better prediction performance for the machine learning approaches, which can distinguish noise levels on different categories of roads.

Traffic noise maps have also been previously developed using top-down approaches. A top-down approach to developing traffic noise maps either breaks down the monitored noises of the environment into its compositional sources, e.g., traffic, or allocates an average noise value to the major routes of the studied area according to the average speed. There have been no certain assessments nor discussions on the contribution of individual sources, i.e., vehicles, in the above-discussed studies. Most of the existing static traffic flow maps have been developed using numerical models supported with limited surveys and/or data collected from the road. Within the literature, the validity of predicting the dynamic spatiotemporal characteristics over the roads has been questioned, see for example the study published by the authors of [29].

Depending on the availability of information and the level of detail required, bottom-up approaches are typically proposed to achieve a detailed and reliable picture of road transport, see for example the studies published by the authors of [30,31]. In bottom-up approaches, the measured noise of the vehicles is accompanied by real-world road data to create a traffic noise map which can address the spatiotemporal features of the roads.

Previous investigators have used either traffic or empirical models when developing traffic noise maps to estimate the average vehicle speed over roads. Traffic models are developed using numerical models as well as utilising the limited volume of road data collected from test cars and/or surveys. There are vigorous debates on the credibility and reliability of the traffic model results, especially regarding their highly detailed spatial and temporal purposes; traffic models provide the annual average speed over arterial routes; hence, the impacts of place and time of travel were missed. Empirical models have also been constructed on the correction factors estimated through hourly monitoring of the total noise across the urban environments, see for example the study published by the authors of [10]. Neither of these models can provide a detailed, dynamic, and real-world picture of road transport across urban environments mainly due to the use of insufficient real-world road data in their estimations.

In this study, we used a bottom-up approach to develop a traffic noise map. We have designed and developed a new-to-the-market, low-cost acoustic sensor pack named ‘Traffic Ear’, which determines the noise level and speed of passing vehicles, as well as their specifications, i.e., vehicle class and engine type. We acknowledge that there is already a wide range of acoustic sensors in the marketplace, so we have provided a summary of the existing acoustic sensors followed by a detailed discussion on the design of ‘Traffic Ear’ in the following section. The rest of this paper is organised as follows: material and methods are discussed in Section 2, where we also provide a literature review on the acoustic sensors and the method for estimating the traffic noise of the studied area. The results of the study are discussed in Section 3, and the main outcomes of this research are discussed and concluded in Section 4.

## 2. Materials and Methods

### 2.1. Acoustic Sensors and Traffic Ear

#### 2.1.1. Literature Review

Traffic monitoring is a prerequisite element of all intelligent transport systems across the globe. Acoustic sensors, which rely on the analysis of sound waves emitted by road vehicles, are promising assets for such purposes. In contrast to the existing and previously established systems, such as radar and lidar sensors, acoustic sensors do not emit any signals; hence, they do not have any side effects on humans, urban monitoring, or communication systems. Several approaches have been proposed in the literature to detect the passing of vehicles using acoustic sensors. For example, Ishida et al. and Uchino et al. used a method that relied on the time difference of arrival (TDOA) of sound signals on two microphones on a sidewalk to detect the vehicles [32,33]. Several other researchers demonstrated the use of a new method based on the analysis of sound intensity signals measured in a two-dimensional space to determine the average speed of the vehicle and the direction of traffic flow [34,35,36]. Other approaches to acoustic vehicle detection have been constructed on machine learning-based techniques, as discussed by Gatto and Forster [37]. To our knowledge, sound intensity analysis was not previously used for detecting the engine type (including diesel, petrol, or hybrid/EV) as well as the class (car, bus, van, HGV, etc.,) of the vehicle. Vehicle class and speeds were usually determined with the aid of image processing (using deep learning algorithms) from video cameras, see for example the study published by the authors of [38].

In this study, we proposed a new approach based on the combination of noise signal analysis, image processing, and machine learning to study the noise of vehicles according to their class, engine type, and speed. Our approach was based on simultaneous visual and acoustic assessments that were supported using the machine learning algorithms.

#### 2.1.2. Traffic Ear

Traffic Ear is the name of the sensor pack that was designed during this study, which was mounted on a roadside structure above the road surface at 5 m from the ground. It attaches to and is powered by a street light column, and works without interrupting the traffic flow, cutting the road surface, or requiring other major infrastructure. Traffic Ear includes an acoustic sensor, a camera, a sensor to measure the ambient concentration of particulate matter (PM), as well as supporting electronic circuitry. It can identify passing vehicles with their engine noise, which is mainly of low frequency, but also contains high frequency harmonics [39]. An image of the Traffic Ear mounted in the city of Sandwell in the West Midlands, UK, is shown in Figure 1.

This acoustic sensor includes a four-channel microphone, as displayed in Figure 2a. We used the beamforming technique to analyse the main noise sources during the pass-by of a vehicle. Beamforming has previously been applied successfully to analyse the noise structure of aeroplanes and trains; meanwhile, only a few research studies have used this technique for traffic noise purposes, see for example the studies published by the authors of [40,41]. We used the beamforming approach in determining the vehicle speed, traffic flow direction, and engine noise through a method developed by Ballesteros et al. and Sarradi et al. [40,41,42]. The basic idea was to focus on an assumed source position and to apply signal processing to the microphone signals.

Essentially, the total intensity of the moving object was measured first, and the source position was then determined using a Kalman filter in a method that was previously proposed by Szwoch and Kotus [36]. The object was then considered by the sensor if its total intensity was found to be higher than the usual noise of road vehicles. Together, with our real-world measurements, it has been shown that the sound wave energy (total) emitted by moving road vehicles is concentrated in the mid-frequency range of 50–200 dB [36]. In order to avoid false-positive recordings, only the objects located in the front area of the sensor with a total length of 85 m were considered. The front area is subdivided into three zones (zones A, B, and C, as displayed in Figure 2b), for which three parts of microphone arrays were allocated for each zone, as indicated in Figure 2a. In zones A and C, which cover 80 m in total (2 × 40 m), the total intensity of the detected object was compared with the estimated background noise, and if it was greater than the detection threshold, this signifies that a vehicle was detected, otherwise the background noise estimate was updated. The sensor then analysed the external tyre noise produced by the tyres of the detected vehicle to determine whether it is approaching or departing, i.e., the direction of traffic flow. A deep learning algorithm was then developed that uses the direction of arrival (DOA) algorithm, which determines the tyre noise as well as the background noise and vehicle speed, in an approach discussed in detail by Szwoch and Kotus [36]. In Zone B, which covers 5 m of the road ahead of the place of the sensor, the engine noise of the vehicle was then analysed.

To enhance the accuracy of the Traffic Ear in terms of its estimated vehicle speed, vehicle counting, and vehicle class detection, we accompanied a camera to the acoustic sensor to simultaneously monitor the moving vehicles and tally up the results. Visual processing of the captured images for vehicle counting, speed estimation, and classification was conducted by many previous investigators, such as the authors of [38,43,44]. In an online object-tracking method (see the study published by the authors of [45]), the vehicle speed was estimated through a certain frame with virtual boundaries, as shown in Figure 2c. We used a convolutional neural network algorithm discussed in detail by Hu et al. [46] to determine the class and speed of the passing vehicles. A deep learning algorithm compares the results of the acoustic and visual parts of the Traffic Ear, which are examined using the acoustic sensor and the camera, respectively. Traffic Ear then stores the result if there is an acceptable agreement (of less than 5% difference) between the vehicle speeds measured using the acoustic sensor and the camera. The block diagram of the designed system for determining the noise and speed of the passing vehicles is shown in Figure 2d. The fuel type of an engine was determined through a deep learning acoustic detection algorithm, which was designed based on the method proposed by Göksu [47]. Traffic Ear was also equipped with a telecommunication part, which permits the storage of the measured valid results in the Cloud and to use them online.

The designed deep learning algorithm also trains according to the background noise, the noise signature of diesel/petrol engines, etc. We mounted Traffic Ear in three locations of the city of Birmingham and the metropolitan borough of Sandwell, UK, before using it for the current study. The deep learning algorithms were then trained for over 1,000,000 measurements. During the training campaign, we also validated the measured intensities using a sound level meter (Tadeto, model SL720), which was able to measure sound levels in the range from 30 to 130 dB with an uncertainty of ±2.0 dB.

### 2.2. The Spatial Scope of the Study and the Locations of the Measurements

The traffic noise map was developed for the West Midlands road network in the UK. The West Midlands had a population of over three million people in 2021, and is the second most populous county in England, after Greater London. The West Midlands county includes the seven metropolitan boroughs of Birmingham, Coventry, Dudley, Sandwell, Wolverhampton, Solihull, and Walsall. The City of Birmingham, with a population of over 1.2 million reported in 2021, is the biggest borough in the West Midlands. The West Midlands and its boroughs are represented in Figure 3a. According to the road traffic data of the Department for Transport (DfT) in the UK [48], in 2021, 28.3 billion vehicle miles of traffic were travelled across the 20.8 thousand miles of roads in the West Midlands, ranking the West Midlands as the region with the fourth highest levels of traffic in Great Britain.

The contribution of each of the boroughs to the population of the West Midlands, the road length, and miles of traffic are reported in Table 1. The dominant contribution of the city of Birmingham to the overall traffic in the West Midlands was observed in this study. Vehicle miles here refer to the total distance travelled by all vehicles during the year [45], while road length is given as the actual length of the roads in the area.

Traffic Ear sensors were mounted in ten locations in the borough of Sandwell, in the West Midlands, between April and October 2022. It should be noted that researchers from the Sandwell Metropolitan Borough Council were involved in this research and therefore arranged for most of the sensors to be installed in the city of Sandwell. The locations of the Traffic Ear devices have been indicated on the West Midlands map (as displayed in Figure 3) by blue push pins. Traffic Ears were attached to the street light columns so that they did not disturb the traffic flow for the measurement period. For the study period, nearly 300,000 valid measurements were recorded.

### 2.3. Vehicle Telematics Data and the Method of GeoSTMUM

Vehicle telematics data, which are collected during the telecommunication between the GPS-connected vehicles and the positioning satellites, can provide a real world and detailed picture of road transport. Vehicle telematics data are mainly collected from the vehicles belonging to drivers who are going to enjoy fairer insurance premiums and so voluntarily share their location data. There are various methods for collecting vehicle telematics data, such as black boxes, driver cell phones, etc., see the review paper of Ghaffarpasand et al. [50]. Recently, Ghaffarpasand and Pope proposed the approach of geospatial and temporal mapping of urban mobility (GeoSTMUM) to convert vehicle telematics (location) data into several urban mobility characteristics, such as the average speed of vehicles, and the percentage of time spent idling, cruising, accelerating, etc. [51]. GeoSTMUM disaggregates road transport in the West Midlands into over 300,000 GeoST segments, and then estimates the urban mobility characteristics over each GeoST segment [52]. GeoST segments are polylines features with a length range of 15–150 m, which cover all features of road networks, such as roundabouts, crossroads, bends, etc. GeoST segments have certain spatial and temporal characteristics. Their spatial characteristics were determined using their geographic coordinates (latitude and longitude). To define the temporal characteristics of the GeoST segments, the annual data was split into 35 time slots, including seven diurnal time slots (00:00–06:59, 07:00–08:59, 09:00–11:59, 12:00–13:59, 14:00–15:59, 16:00–18:59, and 19:00–23:59, respectively) in five days (Mondays, Tuesdays, Fridays, Saturdays, and Sundays, respectively). It was assumed that the traffic behaviour on Wednesdays and Thursdays was similar to Tuesdays. The selected hours of the day were chosen to correspond to weekday ‘early morning hours’, ‘morning rush hours’, ‘morning non-rush hours’, ‘noon rush hours’, ‘afternoon non-rush hours’, ‘evening rush hours’, and ‘evening non-rush hours’, respectively.

In this study, we used the GeoSTMUM approach to estimate the average speed of the vehicles that moved over the West Midlands roads for different time slots of the year 2018. The vehicle telematics data were supplied by the Floow (www.thefloow.com), a telematics UK-based company.

### 2.4. Fleet Composition

A bottom-up approach was used to develop the traffic noise map of the studied area. Fleet composition in terms of the distribution of vehicle classes, i.e., cars, buses, vans, etc., over the road fleet is one of the major requirements to create noise maps of the roads. Traffic Ear can determine the class as well as the fuel type of the monitored vehicles, and we used the estimated fleet composition in our calculations. We used the results of an automatic number plate recognition (ANPR) camera as our reference data to evaluate the performance of the Traffic Ear sensor in determining the fleet composition. The ANPR camera was mounted in a place that was indicated using a red push pin in Figure 3 for two weeks (May–April 2022) to achieve a proven reliable fleet composition of the studied area. ANPR cameras convert the taken picture of the registration number plates (reg numbers) of the passing vehicles into their number digits. The corresponding information of the collected reg numbers was then extracted from the existing driving archives provided by the Driver and Vehicle Licensing Agency (DVLA). From this dataset, we used anonymised statistics (in compliance with the GDPR) of nearly 57,000 records to estimate the fleet composition of the studied area.

### 2.5. Noise Map Development

As discussed by previous investigators, such as the authors of [17], traffic flow speed and its corresponding average speed-based noise levels are the major factors underlying the prediction of the traffic noise maps. In this study, we used Traffic Ear sensors.

Our vehicle noise measurements showed a significantly linear relationship between the average speed of the passing vehicles, within all vehicle classes, and their noise intensity. The linear relationship between the noise intensity and the vehicle speed was also observed in the study of Ref. [10]. The noise intensity of the passing vehicles was estimated in this study using the following equation:(1)$$L_{k}=a_{k}\times v+b_{k}$$
where $L_{k}\;\left(\mathrm{dB}\right)$ and v (km/h) are the noise intensity of vehicle class k and the speed of a passing vehicle, respectively. $a_{k}$ and $b_{k}$ are the constant coefficients which were determined during the linear regression analysis for the vehicle class. We studied five classes of vehicles, which were as follows: petrol cars, diesel cars, vans, buses, and heavy goods vehicles (HGVs).

The results of the regression analysis of the traffic noise measurements have been reported in Table 2. The *p*-values for all the cases examined were less than the significance level of 0.05, which demonstrates a strong linear correlation between the average speed and the noise intensity if accompanied with high values of correlation coefficients.

The average speed of the passing vehicles over the roads was estimated using the GeoSTMUM method. Traffic noises over each GeoST segment were then estimated using the following equation:(2)$$L_{\left(i,j\right)}=\sum_{k}g_{k}\times L_{k,i,j}$$
where $L_{\left(i,j\right)}$ is the average traffic noise (dB) over a GeoST segment, where (i,j) are the spatial and temporal characteristics of the studied GeoST segment, respectively. $g_{k}$ is the contribution of vehicle class k to the total fleet, which was determined using the fleet composition of the studied area. We highlight that $L_{\left(i,j\right)}$ is the average traffic noise on a per-vehicle basis and must be multiplied by the traffic flow activity (the corresponding number of passing vehicles for the studied GeoST segments) if the net amount of traffic noise is desired.

### 2.6. Rush/Non-Rush Hour and Weekday/Weekend Effects Analysis

Here, traffic noise was estimated for the GeoST segments covering the studied road network spatially and temporally in 35 time slots for the year 2018. The rush/non-rush hours and weekday/weekend effects were then analysed by calculating the relative difference between a pair of corresponding cases over each GeoST segment. In other words, for example, the relative difference between the rush hour and non-rush hour traffic noise on a particular road section (GeoST segment) represents the rush hour/non-rush hour effects on the traffic noise of the GeoST segment under study.

The 5% trimmed mean was then used to provide an average of the relative differences examined. The trimmed mean is defined a statistical measure of central tendency that involves determining the mean after discarding certain parts of a probability distribution or sample at the high and low extremes. In a 5% trimmed mean, the lowest 5% and highest 5% of the data are excluded, and the mean is calculated from the remaining 90% of the data points. We used the 5% trimmed mean to exclude the likely outliers.

In order to analyse the weekday/weekend effect, the weighted average was used to ensure that the different weights for the weekdays and weekends (in terms of the number of weekdays and weekends) were considered.

## 3. Results and Discussions

### 3.1. Fleet Composition

The performance of the Traffic Ear sensor in terms of fleet composition estimation was assessed against the results obtained through the ANPR campaign reported in Table 3. Both techniques recorded the dominant contribution of cars (78–82%) to the total fleet. This complies with the UK fleet statistics that have been estimated for the areas outside of London [53]. Fleet composition by fuel type investigated by Traffic Ear and ANPR camera is shown in Figure 4a,b. Figure 4 reveals that the Traffic Ear sensor was able to determine the class of vehicles with a degree of uncertainty between 1–4%, which was deemed an acceptable level of agreement. The class of 5% of the passing vehicles was not able to be determined using either of these methods. However, it was noted that the offline results of the ANPR camera were extracted after a long and complicated administrative procedure through stringent legal restrictions. In contrast, Traffic Ear determined the class of the vehicle in real time. Traffic Ear provides an online picture of road transport in terms of the number and class of moving vehicles. However, a few partial discrepancies were observed in the estimation of the fuel type of the passing vehicles. Whilst Traffic Ear estimated an equal contribution of petrol and diesel cars, ANPR results indicated a higher proportion for petrol cars. It is worthwhile to note that Traffic Ear determines the vehicle class through a mutual visual-acoustic assessment, while the fuel type was estimated through an intelligent acoustic signature-matching procedure. Hence, it is expected that the uncertainty in the determination of the fuel type will decrease through further algorithmic training over time. Traffic Ear can also detect hybrid/electric vehicles; a car is specified as hybrid/electric if it is not specified as either petrol or diesel. However, given the small share of hybrid/electric vehicles in the fleet (at the date of this study), more training will be needed to provide reliable outputs for these vehicles. Currently, an uncertainty level of 5–8% in fuel type determination was achieved in the current version of Traffic Ear. The fleet composition was estimated using the Traffic Ear for different locations in the study area; therefore, we used that for the development of the traffic noise map.

### 3.2. Traffic Noise Assessment on a Per-Vehicle Basis

The traffic noise maps of the studied area for different time slots were developed using the method discussed in Section 2.5. The results were provided here through considering the results of the corresponding GeoST segments. These results provide an estimation of the noise emitted from an average vehicle passing through the road network. It does not currently provide a total noise intensity of the whole fleet, which would be the total integrated noise of all moving vehicles passing over a specific road segment per unit of time. Estimating the road noise intensity, which requires additional information, such as the traffic flow and road occupancy, is a future direction of this research.

The annual distribution of average vehicle noise (on a per-vehicle basis) across the West Midlands road network is shown in Figure 5a. The high level of average vehicle noise on arterial roads (motorways and trunk roads) was determined to be due to the increased speeds observed on these roads. The probability distribution function (PDF) of the annual average speed of the different roads studied is shown in Figure 5b. Vehicles have higher speeds on motorways, meaning therefore that arterial roads are the hotspots of traffic engine noise. Motorways, trunk roads, primary roads, and secondary roads contributed 8%, 34%, 23%, and 35%, respectively; hence, 70% of the studied road types were either secondary or trunk roads. In Figure 5c, we show the PDF of the traffic noise factor (dB/m) as the annual average traffic noise per average vehicle over each GeoST segment per segment length on the studied road types. The PDF profile of the traffic noise factor in motorways exhibited a sharper peak than the profile for the other road types. The median value of the PDF of the traffic noise factor for the motorways, primary, secondary, and trunk roads was 0.47 dB/m, 0.38 dB/m, 0.35 dB/m, and 0.40 dB/m, respectively. This was attributed to the higher average speed of vehicles on motorways compared with that on other roads.

### 3.3. Spatiotemporal Distribution of the Traffic Noise

The probability distribution function (PDF) of the average vehicle noise level over the GeoST segments with different spatial and temporal characteristics has been studied in this section. We remind the reader that the results presented here are the average vehicle noise levels, and not the total noise intensity, across the roads studied. Figure 6 shows the PDF of traffic noise for different hours over a day across the studied roads. It reveals that GeoST segments placed in the motorways have higher levels of traffic noise than those placed in the other studied roads. The major peak of the PDF profiles in the motorway GeoST segments was around 60 dB, while the dominant contribution of the GeoST segments in the other roads displayed traffic noises smaller than 40 dB, respectively. The hourly traffic noise variation noted in the motorways was higher than that observed in the other studied roads. A considerable reduction was observed in the PDF profiles of traffic noise for the morning and evening rush hours in the motorways compared with the non-rush hour profiles. However, in all the studied roads, the PDF profile of traffic noise was shifted to higher values when moving from rush to non-rush hours. These variations correlated with the level of congestion and hence the average vehicle speed over the different hours of the day.

The PDF of traffic noise over the GeoST segments for the different days of the week has been represented in Figure 7. Solid, dashed, and dotted lines correspond to morning rush hours (07:00–09:00), evening rush hours (16:00–19:00), and non-rush hours (19:00–23:00), respectively, while the red two-dashed line represents the noon-rush hours (12:00–15:00). A significant variation was observed in the PDF profile of the motorway GeoST segments. Furthermore, the PDF profiles (of all road types) displayed shorter peaks in the midday rush hour on the weekends compared to the other times examined.

On working days, the major peak of the PDF profile of traffic noise was sharper under non-rush hours than that observed for non-rush hours. It also formed at higher traffic noise values compared with that under non-rush hours. At weekends, and especially on the motorways, the PDF profile of the traffic noise during the morning rush hour period was sharper, with higher traffic noise levels observed compared to the rest of the day.

### 3.4. Rush/Non-Rush Hour and Weekday/Weekend Effects

As previously mentioned in Section 2.6, we parameterized the rush/non-rush hour and weekday/weekend effects by analysing the relative difference between the pair of corresponding GeoST segments for each case. The relative difference between the average vehicle noise during the rush hours and non-rush hours, and between the weekdays and weekends, have been reported in Table 4. The average vehicle noise during rush hours was smaller than that observed under non-rush hours by 18%, 9%, 10%, and 10% for driving over motorways, secondary, primary, and trunk roads, respectively. We also estimated the weighted average according to the contribution of the studied road types; 24.4%, 36.8%, 33.5%, and 5.2% of the studied roads were primary, secondary, trunk roads, and motorways, respectively. Table 4 shows that rush hours reduced the average traffic noise levels per vehicle on the studied roads with a weighted average of 8.4%. The effect of rush/non-rush hours on the motorways was higher than that on the other studied roads. This was determined to be likely due to the wider range of speeds that are allowable on the motorways. The weekday/weekend effect was smaller than the rush/non-rush hour effect, whereby traffic noise on the weekdays was smaller than that at the weekends with an average of 5%.

## 4. Summary, Conclusions, and Future Research Directions

Traffic noise inventories in urban areas have faced several technical challenges, such as the determination of noise from passing vehicles according to their characteristics, the use of traffic flow maps with a low spatial resolution, the lack of temporal assessments, etc.

In this study, we proposed a new bottom-up approach to developing traffic noise maps within urban environments using Traffic Ear, a new-to-the-market sensor pack. Traffic Ear employs machine learning algorithms, using traffic noise, to disaggregate passing vehicles based on their class, speed, and engine type. A major benefit of the Traffic Ear is that it can be mounted on existing street furniture e.g., a lamp post, and can listen to the traffic and determine the noise intensity of passing vehicle engines without interrupting the traffic flow.

Traffic Ear was mounted in ten urban locations from April to October 2022 to create a reliable traffic noise dataset and used the beamforming technique to determine the traffic flow direction, vehicle speed, and engine noise. The Traffic Ear sensor also incorporates a camera that can be used to increase the credibility and reliability of the results. The results obtained from Traffic Ear in terms of the fleet composition and fuel type determination of the passing vehicles were compared with the results of an automatic number plate recognition (ANPR) camera, which was installed in the same location. The results revealed that Traffic Ear determined the fleet composition with a small level of uncertainty (<4%).

We analysed the variation between the traffic noise and the speed of the passing vehicles and found significantly linear relationships for different vehicle subsets. We then used the newly developed method of geospatial and temporal mapping of urban mobility (GeoSTMUM) to estimate the average speed of the passing vehicles over the entire urban environment with high spatial and temporal resolutions. GeoSTMUM uses vehicle telematics (location) data to estimate the urban mobility characteristics, e.g., average vehicle speed, over extensively detailed geospatial and temporal frameworks. This case study was set in the West Midlands, which is the second most populated county after Greater London in the UK. The West Midlands traffic noise map was created from a traffic noise dataset that was derived from the Traffic Ear campaigns (undertaken in the boroughs of Sandwell and Birmingham, West Midlands, UK), fleet compositions estimated using an ANPR camera, and the outputs from the GeoSTMUM methodology. A dominant contribution (almost 70%) of the studied roads in the West Midlands are trunk roads and secondary roads, while motorways have the highest average vehicle noise levels. The median of the PDF profile of the traffic noise factor (dB/m) on the motorways was 19.0%, 25.5%, and 15.0% higher than on the primary, secondary, and trunk roads, respectively, in line with the measured traffic speeds.

Analysis of the traffic noise levels (on a per-vehicle basis) revealed that urban areas have a lower average vehicle noise per vehicle during rush hours and weekends than during off-peak hours and weekdays, respectively. However, it has been shown that the rush/non-rush hour effect on the traffic noise was greater on the motorways than on the other roads. A future research direction for the current study may be to incorporate additional information, such as the traffic flow and road occupancy, to estimate the integrated traffic noise intensity.

This research introduced the Traffic Ear as a low-cost sensor package that can be deployed as a network at many points in the urban environment, providing an online picture of urban mobility. It can telecommunicate geospatial data with the cloud, and thus can be processed and used on the Internet of Things (IoT) platforms. Such cheap sensors can have wide applications in urban digital twins, where data flow between the physical and digital twins needs to be established. The future direction of this research is the feasibility study of the wide applications of such cheap sensors in advanced technologies and paradigms, such as the IoT and digital twins.

## Figures and Tables

**Figure 1 sensors-23-06964-f001:**
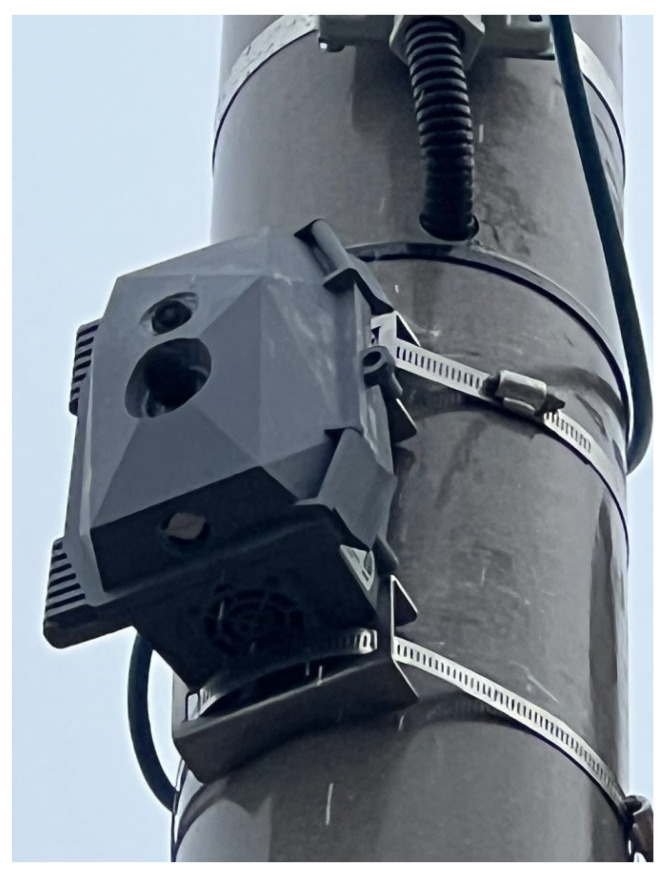
An image of Traffic Ear mounted on a street column in the metropolitan borough of Sandwell, West Midlands, UK.

**Figure 2 sensors-23-06964-f002:**
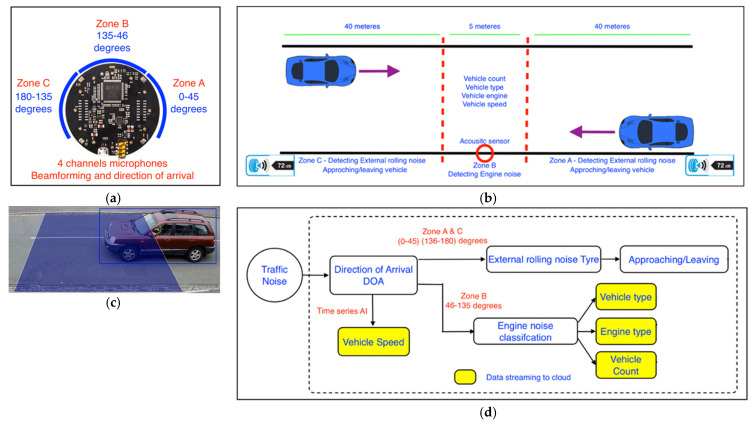
(**a**) The cross picture of the four-channel microphone used in the Traffic Ear; (**b**) the front zones considered by the Traffic Ear; (**c**) the virtual frame boundaries that were captured using the computer vision camera of Traffic Ear; and (**d**) the block diagram of the system designed within the Traffic Ear.

**Figure 3 sensors-23-06964-f003:**
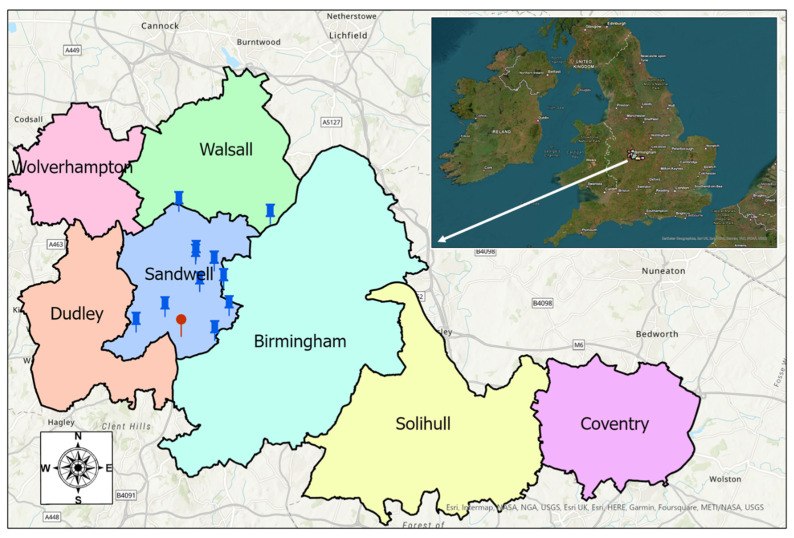
West Midlands boroughs, the locations of Traffic Ear sensors and ANPR cameras are indicated with blue and red push pins, respectively.

**Figure 4 sensors-23-06964-f004:**
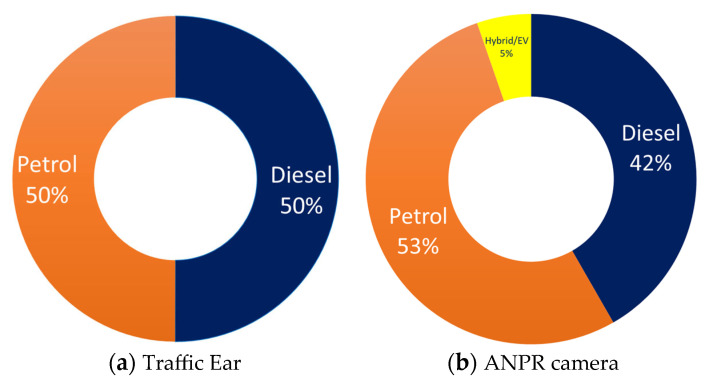
The contribution of diesel and petrol cars estimated using the (**a**) Traffic Ear sensor and (**b**) ANPR camera.

**Figure 5 sensors-23-06964-f005:**
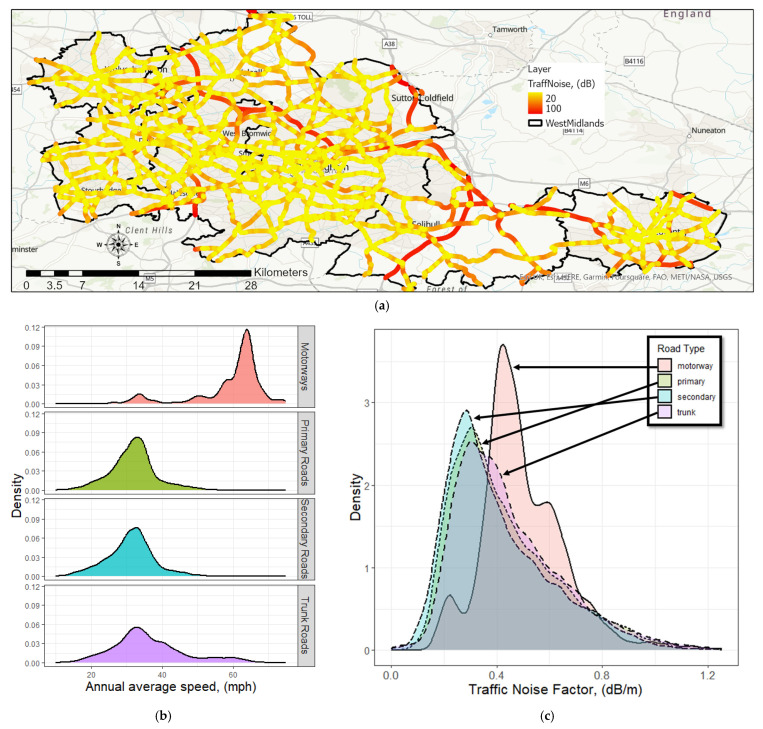
(**a**) Average vehicle noise map (per-vehicle basis) of the West Midlands for the year 2018; (**b**) the probability distribution function (PDF) of the annual average speed over the road studied; and the (**c**) PDF of the traffic noise factor as the annual average traffic noise per vehicle of each GeoST segment per segment length.

**Figure 6 sensors-23-06964-f006:**
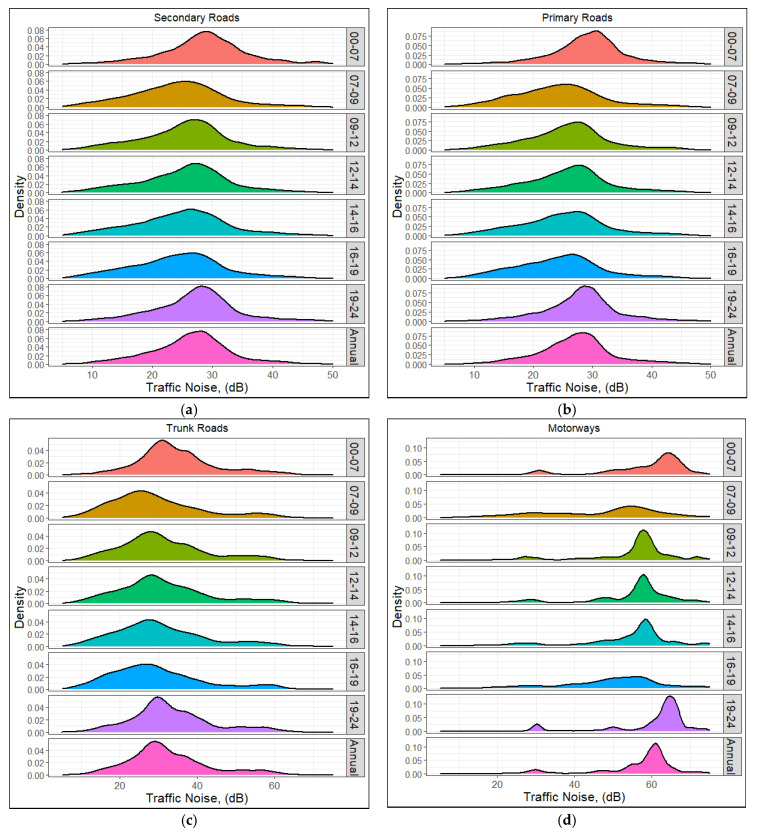
The hourly variation of the traffic noise across the (**a**) secondary roads, (**b**) primary roads, (**c**) trunk roads, and (**d**) motorways of the West Midlands for the year 2018.

**Figure 7 sensors-23-06964-f007:**
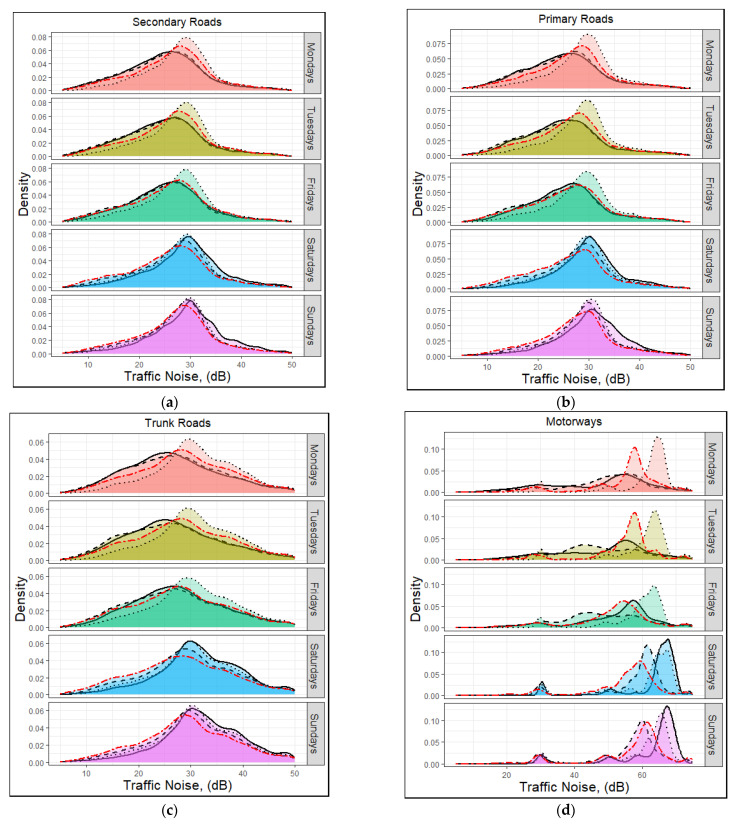
Diurnal variation of the traffic noise across the (**a**) secondary roads, (**b**) primary roads, (**c**) trunk roads, and (**d**) motorways of the West Midlands for the year 2018. Note that the *x*-axis scale is different for the motorways compared to the other road types. Solid, dashed, and dotted lines correspond to morning rush hours (07:00–09:00), evening rush hours (16:00–19:00), and non-rush hours (19:00–23:00), respectively.

**Table 1 sensors-23-06964-t001:** The contribution (%) of the West Midlands boroughs to the population, road length, and vehicle miles across the West Midlands. The statistics are for the year 2021 and were extracted from Refs. [48,49].

Borough	Birmingham	Sandwell	Walsall	Wolverhampton	Solihull	Dudley	Coventry
Population (%)	39	12	10	9	7	11	12
Road length (%)	33	12	11	10	11	12	11
Vehicle miles (%)	35	13	10	7	14	10	11

**Table 2 sensors-23-06964-t002:** Results of the regression analysis of the linear relationship between the average speed and the noise intensity.

Vehicle Class	A (dB h/km)	b (dB)	R-Square	*p*-Value
Petrol cars	1.45	−5.45	0.87	<2.2 × 10^−16^
Diesel cars	1.45	−5.47	0.87	<2.2 × 10^−16^
Vans	1.3	−1.15	0.77	<2.2 × 10^−16^
Buses	1.4	−5.05	0.85	<2.2 × 10^−16^
HGVs	1.3	−4.4	0.86	<2.2 × 10^−16^

**Table 3 sensors-23-06964-t003:** The Traffic Ear and ANPR camera-estimated contributions (%) of different vehicle subsets to the fleet composition of the study area.

	Cars	Vans	Buses	HGVs	NA
Traffic Ear	78	12	2	3	5
ANPR camera	82	11	1	1	5

**Table 4 sensors-23-06964-t004:** Trimmed mean of the relative difference between the traffic noise over a pair of corresponding GeoST segments (see Section 2.6). * The weighted average is the average of the different road types weighted with the occurrence in the whole of the West Midlands.

	Motorways	Secondary Roads	Primary Roads	Trunk Roads	Weighted Average *
Rush/non-rush hour effect	18%	9%	10.3%	10%	8.4%
Weekday/weekend effect	5.3%	4.5%	5.4%	4.7%	4.8%

## Data Availability

Individual vehicle data are not available due to GDPR compliance. We provide the parameterizations of the PDFs shown in the Appendix A.

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
