# Peer review of "Traffic Noise Assessment Using Intelligent Acoustic Sensors (Traffic Ear) and Vehicle Telematics Data"

_sensors, 2023, doi:10.3390/s23156964_

Round 1

Reviewer 1 Report

The paper presents the structure of the Traffic Ear and the results that were obtained by using this sensor on different locations o United Kingdom to record and inference information about the traffic road.

The presented method that is embedded in Traffic Ear generates noise of the vehicles depending of the following results: the engine type, the class and the speed of the vehicles. For this purpose a hardware structure that use an array of four microphones and a camera is used and also a software part that includes signal processing techniques an machine learning algorithms.

The authors claims that this method is a new one but they did not present details about the implementation. They only cites some references that have already implement some parts from their implementation.

For instance there are no details regarding how microphone data and camera data are used. On the other hand, the Traffic Ear devices are mounted on street columns. What was the method to transfer the results to the user? Do these devices have an electronic part so they can be seen as Internet of thing devices?

Finally an observation about the typing: on the fourth page, the seventh row from the end, it should be Zone B instead Zone C.

Reviewer 2 Report

Overall Comments: 

Overall, this is a well-written paper about the implementation of many Traffic Ear monitors across a Burrough in West Midlands to better assess noise from traffic at different times of the day, days of the week, and by vehicle fleet type and engine type. I enjoyed reading about this tech and look forward to watching it progress. My specific comments are below but generally, there are some methodological details missing, the figures that include pie charts should be reworked into a more digestible format, and the summary/conclusion section should tie a bit more directly back to the thesis of the paper/product. Since the authors start the paper with “Here we introduce traffic ear” they should include more details about it and more about the validation of its other features, particularly noise monitoring.  

I selected "accept after minor revisions" because I think that all of my comments are in fact minor and if addressed directly do not need re-review but rather can be assessed for completeness by the editor.

Abstract: 

- Why did the measurements show a reduction in traffic noise over the whole of the study of 8%?

- Should the last sentence say “almost always”? 

Introduction: 

- Consider rewording this first sentence. Although it is a part of the backbone of the economy, I would argue that it is not the backbone of social well-being or sustainable development. This first sentence doesn’t set the tone properly for the introduction in my opinion.  

- Rather than writing Ref. Just say the first author's last name et al throughout. Or phrase the sentence in a way that it can just have the reference at the end of it.  

- I would suggest pulling a recent systematic review about the health effects of noise rather than two rather specific studies to highlight the “wide body of research” showing the ill effects of noise.  

Materials and Methods: 

- What is the limit to the Distance of Detection? Are we to assume that a car is not detected until they are 40 or fewer meters from the sensor? More details about the sensitivity and range of this device need to be included. Also, there is no information about the validation of the noise measurements instead this paper focuses on validating the fleet and engine determination while still presenting the noise values. Have they been validated elsewhere? If so, please summarize. 

- Is there a speed limit? Or in other words, if a car was going 100 mph would the sensor be fast enough to detect it? Similarly, is there a low end to the speed where the sensor fails or is error-prone? 

- Why go into all the details about Birmingham when Sandwell was where most of the sensors were set up? It would be more useful to discuss why Sandwell was chosen after giving an overview of the area generally.  

Figure 3: Is it correct to conclude that only one ANPR camera was set up since there is only one red push pin on the map? These pie charts are not very easy to read or gather information from particularly since the colors don’t mean anything other than labeling the borough throughout. I think this data would be more digestible as a table. Generally, the point is that Birmingham is by part the largest and more contributing, while Sandwell is second but contributes quite a bit less. 

- Why was ANPR only in one location to verify the automatic fleet composition determination? 

- Was any modeling undertaken for motorcycles? Could they be contributing to the 5% not identifiable by either method? Motorcycles are noisy.  

Results and Discussions: 

- It’s apparent that the section about rush/non-rush hour and weekday/weekend effects on analysis is supposed to speak to what was done to account for the differences but it is not written very clearly. Please clear up this paragraph.  

- Does the PM monitoring on the Traffic Ear contribute to the signatures used to determine the fuel type of the passing cars at all? It was mentioned as a feature in passing but never discussed again.

Figure 4: Similar to Figure 3 the pie charts are not the most helpful way to present this information.  

Figure 5: Similar to Figure 3 the pie charts are not the most helpful way to present this information. 

- It would be helpful for the reader if you could summarize the typical speeds on the different road types to contextualize the noise difference a bit better. 

- For Figure 7 and the related data, what about the time between 9 am and 4 pm? Seems like there would be significant delivery and larger truck/van traffic during this time if my experiences in the US are at all similar. Why not present some time period within that time as another non-rush hour period? 

- The first sentence of the last paragraph in section 3.3. is written incorrectly.  

Figure 6 & Figure 7: Please make sure that the scales for all of these graphs on both x- and y-axes are the same so that the reader can easily see the differences by roadway that are described in the text. I know that you have a note in Figure 7 that the scales are different but I think for clarity and reduction of potential misinterpretation by readers they should all be the same. Why aren’t Wednesday and Thursday presented in these graphs? If it’s because they look like Monday and Tuesday, please say that and maybe consider averaging across the days and presenting them as Monday-Thursday. Otherwise, Figures 6 and 7 are very helpful graphs 

Table 2: What do you mean by “trimmed mean” in this case? Please describe. Since all of these values are positive (I.e., relative in this case) it doesn’t highlight the directional difference between the two comparators. I would suggest either making the values directional or reminding the reader in the table title what the outcomes were. 

Summary and Conclusions: 

- I think the section is good but I think it should tie a bit better back to the gaps in noise monitoring and mapping that this device was/is trying to fill and how it succeeded or didn’t and what challenges lie ahead.

The paper is well-written. No issues with the English language. Minor grammatical and clerical errors. 

Round 2

Reviewer 1 Report

The authors answered to my questions. I am satisfied with the answers.